

# Air quality modelling in the summer over the Eastern Mediterranean using WRF/Chem: Chemistry and aerosol mechanisms intercomparison

George K. Georgiou[1], Theodoros Christoudias[2], Yiannis Proestos[1], Jonilda Kushta[1], Panos Hadjinicolaou[1], and Jos Lelieveld[3,1]

[1]Energy, Environment and Water Research Center, The Cyprus Institute, Nicosia Cyprus
[2]Computation-based Science and Technology Research Centre (CaSToRC), The Cyprus Institute, Nicosia, Cyprus
[3]Air Chemistry Department, Max Planck Institute for Chemistry, Mainz, Germany

*Correspondence to:* George K. Georgiou (g.georgiou@cyi.ac.cy)

**Abstract.** We employ the WRF/Chem model to study summertime air pollution, the intense photochemical activity and their impact on air quality over the Eastern Mediterranean. We utilize three nested domains with horizontal resolution of 80km-16km-4km, with the finest grid focusing on the island of Cyprus, where the CYPHEX campaign took place in July, 2014. Anthropogenic emissions are based on the EDGAR HTAP global emission inventory, while dust and biogenic emissions are calculated online. Three simulations utilizing the CBMZ-MOSAIC, MOZART-MOSAIC, and RADM2-MADE/SORGAM gas-phase and aerosol mechanisms are performed. The results are compared with measurements from a dense observational network of 14 ground stations in Cyprus. The model simulates $T_{2m}$, $P_{surf}$, and $WD_{10m}$ accurately, with minor differences in $WS_{10m}$ between model and observations at coastal and mountainous stations attributed to limitations in the representation of the complex topography in the model. It is shown that the south-eastern part of Cyprus is mostly affected by emissions from within the island, under the dominant (60%) westerly flow during summertime. Clean maritime air from the Mediterranean can reduce concentrations of local air pollutants over the region during westerlies. Ozone concentrations are overestimated by all three mechanisms ($9\% \leq NMB \leq 23\%$) with the smaller mean bias (4.25 ppbV) obtained by the RADM2-MADE/SORGAM mechanism. Differences in ozone concentrations can be attributed to the VOC treatment by the three mechanisms. The diurnal variability of pollution and ozone precursors is not captured (hourly correlation coefficients for $O_3 \leq 0.29$). This might be attributed to the underestimation of $NO_x$ concentrations by up to 50%. For the fine particulate matter ($PM_{2.5}$), the lowest mean bias (9 $\mu gm^{-3}$) is obtained with the RADM2-MADE/SORGAM mechanism, with overestimates in sulphate and ammonium aerosols. Overestimation of sulphate aerosols by the RADM2-MADE/SORGAM mechanism may be linked to the heterogeneous $SO_2$ cloud oxidation. The MOSAIC aerosol mechanism overestimates $PM_{2.5}$ concentrations by up to 22 $\mu gm^{-3}$ due to a more pronounced dust component compared to the other two mechanisms, mostly influenced by the dust inflow from the global model. We conclude that all three mechanisms are very sensitive to boundary conditions from the global model for both gas-phase and aerosols pollutants, in particular dust and ozone.



## 1   Introduction

Many years of intense population growth have rendered the Eastern Mediterranean and the Middle East (EMME) region into a very densely populated area with more than 350 million inhabitants over an area with a 2,000 km radius. Strong industrialisation and a lack of air pollution policy in the countries in the region have resulted, in recent decades, in an increase of anthropogenic emissions to the atmosphere. Compared to other regions in the Northern Hemisphere, background concentrations of important trace gases and aerosols over the EMME region are very high (Lelieveld et al., 2002), whilst the Mediterranean basin is found to be the region with the highest background ozone ($O_3$) levels in Europe. Several locations in the Middle East are characterised by much higher nitrogen dioxide ($NO_2$) column densities than major cities in Europe (Lelieveld et al., 2009).

The Eastern Mediterranean atmospheric $O_3$ concentrations are characterised by seasonal variability with the maxima observed during the summer (Kouvarakis et al., 2000; Gerasopoulos et al., 2005; Kalabokas et al., 2008; Kleanthous et al., 2014) due to intense photochemical activity and the prevailing meteorological conditions. The collocation of the Azores high pressure system and the Asian monsoon low pressure regime to the east causes northerly circulation over the Aegean Sea that sheers to north-westerly over the Eastern Mediterranean. As a result, the EMME region is affected by near-surface transport of polluted air masses from various distance sources such as Near-Asia, East and Central Europe (Lelieveld et al., 2002; Gerasopoulos et al., 2005; Ladstätter-Weißenmayer et al., 2007; Kalabokas et al., 2008; Kanakidou et al., 2011). Kleanthous et al. (2014) reported that long-range transport (LRT) has important impacts on the air quality over the island of Cyprus and it is directly linked to high $O_3$ levels. Local precursor emissions such as nitrogen oxide (NO) and carbon monoxide (CO) have been found to account only for 6% of the observed $O_3$ levels.

Downward transport from the upper troposphere and lower stratosphere associated with enhanced subsidence and limited horizontal divergence has been found to be another important mechanism, which increases the already elevated $O_3$ concentrations over the EMME region (Zanis et al., 2014). LRT also enhances carbon monoxide (CO) surface concentrations, with 60% to 80% of the boundary-layer CO over the Mediterranean attributed to polluted air masses originating from western and eastern Europe, while the Eastern Mediterranean is mainly affected by emissions from Ukraine and Russia (Lelieveld et al., 2002).

During the second phase of the Air Quality Model Evaluation International Initiative (AQMEII), the nine working groups using the Weather Research and Forecasting model coupled with chemistry (WRF/Chem) operationally reported an overall underestimation of the annual surface ozone ($O_3$) levels reaching up to 18% over Europe and 22% over North America (Im et al., 2015a) with autumn overestimation and winter underestimation. The meteorological and chemical configurations of the different groups were found to have a considerable effect on simulated $O_3$ levels. Model performance was strongly influenced from the boundary conditions, especially during autumn and winter. Regarding particulate matter 2.5 micrometer or less in diameter ($PM_{2.5}$) concentrations, large overestimations over Europe were reported (Im et al., 2015b). Tuccella et al. (2012) compared WRF/Chem model output against ground-based observations over the European domain for the year 2007 with time-invariant boundary conditions. The model simulated temperature satisfactorily with a small negative bias, but wind speed was highly overpredicted. $O_3$ daily maxima were underestimated, while mean $O_3$ concentrations during spring (fall) were



underestimated (overestimated). Ritter et al. (2013) applied the model over a Swiss domain for two years on a 2km horizontal resolution. The model reproduced well temperature and solar radiation, but failed to capture short-term peaks in pollutant concentrations for several days.

In the literature various gas-phase chemistry and aerosol mechanisms have shown different behaviour in terms of predict-
ing the atmospheric concentrations of pollutants over specific regions. Gupta and Mohan (2015) compared the Carbon Bond Mechanism (CBM-Z) and the Regional Atmospheric Chemical Model (RACM) gas-phase chemistry mechanisms over the mega city of Delhi, India on a horizontal grid resolution of 10 km for the innermost model domain. Results showed that both mechanisms tend to overestimate $O_3$ concentrations. It was noted that the use of a finer grid resolution may improve the overall model performance. Mar et al. (2016) evaluated the performance of the Regional Acid Deposition Model (RADM2) and
MOZART-4 gas-phase chemistry mechanisms on a horizontal grid resolution of 45km over Europe. Simulated $O_3$ consecrations by MOZART-4 were found to be up to $20 \mu g m^{-3}$ higher than RADM2 during the summer due to a higher photochemical $O_3$ production rate. On the other hand, RADM2 showed a negative bias for the whole year, while both mechanisms slightly underestimated nitrogen oxide ($NO_x$) concentrations. Knote et al. (2014) performed box-model intercomparison of several formulations for tropospheric gas-phase chemistry under idealized meteorological conditions in the framework of the second
phase of AQMEII. They found significant variabilities in the prediction of gaseous pollutants and key radicals and they highlight that the choice of gas-phase mechanism is a crucial component in modelling studies. Balzarini et al. (2015) showed that predicted total PM mass concentrations as well as aerosol subcomponents vary between the MADE/SORGAM and MOSAIC aerosol mechanisms. Differences were attributed to the approach each mechanism uses to simulate the aerosol size distribution (modal or sectional bin) and the gas-phase chemistry mechanisms these are coupled with in the WRF/Chem model since they
affect the concentrations of aerosol precursors.

A very limited number of studies have dealt with on-line air quality modelling over the EMME region, with apparent limitations. Safieddine et al. (2014) employed the WRF/Chem model to study the tropospheric $O_3$ over the Mediterranean during the summer season on a horizontal grid resolution of 50km. The coarse model horizontal grid resolution was proposed by the authors as a possible reason for model biases in their study. Other studies in the region that utilize coupled meteorological
and chemistry models are usually short term. For example, Bossioli et al. (2016) carried out WRFC/Chem simulations for a limited time period focusing on the contribution of biomass burning on PM levels. However they reported an increase in $O_3$ levels by 50% when boundary conditions from the MOZART-4 global model were used. Kushta et al. (2014) highlighted the importance of natural aerosols when simulating the photochemical state of the atmosphere during a dust episode in April 2004.

In this study we employ and intercompare three coupled gas-phase chemistry and aerosol mechanisms to study the long-
range transport of air pollutants and the intense photochemical activity over the Eastern Mediterranean with focus on the island of Cyprus, over the summer period, using high temporal and spatial resolution down to 4 km. During July 2014, the CYprus Photochemical Experiment (CYPHEX) Campaign took place near Ineia, Paphos, a background measurement site on the western coast of Cyprus, to investigate the photochemistry and air mass transport of the eastern Mediterranean, providing us with an extensive observation data set.



The paper is structured as follows: In Section 2 we briefly describe the three gas-phase chemistry and aerosols mechanisms used in the simulations, the basic model configuration including the model domains, the common parameterizations and the emission data used. In Section 3 we present the results from sensitivity tests dealing with the effects of boundary conditions on the concentrations of gas-phase pollutants and aerosols (Sec. 3.1). We examine the ability of the model to predict the basic

meteorological parameters (Sec. 3.2), the concentrations of gas-phase pollutants (Sec. 3.3) and fine particulate matter (Sec. 3.4). Our conclusions are given in Sec. 4.

## 2 WRF/Chem model and observations

### 2.1 Gas-phase chemistry and aerosol mechanisms

The Weather Research and Forecasting (WRF) model is a state-of-the-art regional meteorological model. Various gas-phase

chemistry and aerosol mechanisms have been implemented into the WRF model, creating the on-line WRF/Chem model (Grell et al., 2005). In this study, we employ WRF/Chem version 3.61 with three widely-used gas-phase chemistry and two aerosol mechanisms to simulate air quality over the Eastern Mediterranean:

**CBMZ-MOSAIC (CM)** The lumped CBM-Z chemical mechanism (Zaveri and Peters, 1999) is based on the Carbon Bond Mechanism (CBM-IV) developed by Gery et al. (1989). The Carbon Bond Mechanism includes 73 chemical species

and 237 reactions. CBM-Z is coupled with the Model for Simulating Aerosol Interactions and Chemistry (MOSAIC) developed by Zaveri et al. (2008). MOSAIC uses a sectional bin approach for the representation of the aerosol size distribution. In the WRF/Chem model the user can choose between four and eight size bins which are defined by their lower and upper dry particle diameters. In both cases, only one bin is dedicated to aerosols with diameter between 2.5 and 10 $\mu m$. Therefore, when four aerosol bins are used, three bins are dedicated to aerosols less than 2.5 $\mu m$ in diameter

and when eight aerosol bins are used, seven bins are dedicated to aerosols with diameters within this range. Since this study focuses on the total $PM_{2.5}$ mass concentrations and not on detailed aerosol microphysics or effects on clouds, it is sufficient to use the four-bin option to reduce computational complexity.

**MOZART-MOSAIC (MM)** The MOZART gas chemical mechanism, developed by Emmons et al. (2010), is also used coupled with the MOSAIC aerosol scheme. It includes 85 chemical species and 196 reactions and is consistent with the

chemistry used in the global model that provides the chemical boundary conditions for our simulations. The MOZART mechanism has been widely used with WRF/Chem for simulations outside Europe but only a limited number of studies have applied it over the European domain.

**RADM2-MADE/SORGAM (RMS)** The second generation Regional Acid Deposition Model (RADM2) chemical mechanism for regional air quality modelling (Stockwell et al., 1990), includes 59 chemical species and 157 reactions. RADM2

is a widely used mechanism over the European domain and it is coupled with the Modal Aerosol Dynamics for Europe (MADE) (Ackermann et al., 1998). MADE uses a modal approach for aerosol treatment and is coupled with the Secondary Organic Aerosol Model (SORGAM) (Schell et al., 2001). SORGAM is capable of simulating secondary organic




**Table 1.** Physics options used, common in all simulations

| Atmospheric Process | Scheme |
| --- | --- |
| Cloud microphysics | Morrison double moment (Morrison et al., 2005) |
| Cumulus parametrization | Grell 3D (Grell, 1993, 2002) |
| Land-surface physics | Noah Land Surface Model (Chen and Dudhia, 2001) |
| Longwave radiation | RRTM scheme (Mlawer et al., 1997) |
| Photolysis | Fast-J Photolysis |
| Planetary Boundary Layer | Yonsei University PBL (Hong et al., 2006) |
| Shortwave radiation | RRTM scheme (Mlawer et al., 1997) |

aerosol (SOA) formation including the production of low-volatility products and their subsequent gas/particle partitioning.

## 2.2 Model configuration

All our simulations are conducted with the same model physics configuration (Table 1) to facilitate intercomparison. We
modified the WRF/Chem v3.6.1 code to take into account dust particles in the accumulation size mode (0.1 $\mu$gm$^{-3}$ ≤ particle
size ≤ 2.5 $\mu$gm$^{-3}$) for the calculation of the total PM mass concentration in the RMS mechanism. Three nested domains
are used, as shown in Figure 1 (left panel). The outermost domain with a horizontal grid resolution of 80km extends from
16° to 4° north and from 10° west to 50° east in order to include a large part of Europe and the Black Sea region which have a
significant contribution to the pollution of air masses that reach the EMME region, as well as a large part of the Sahara and
Middle East deserts in order to utilize the dust emission schemes included in the WRF/Chem model. The second domain with
a horizontal grid resolution of 16 km is located over the Levantine Basin, including all the surrounding major urban centres.
The third innermost domain is located over the island of Cyprus (Figure 1, right panel) with a horizontal grid resolution of 4km
allowing for a more accurate representation of the state of the atmosphere over the complex terrain of the island close to the
surface observation stations. The WRF/Chem model uses a terrain-following hydrostatic-pressure vertical coordinate system.
In our study 29 layers are used from the surface up to 50 hPa. The first layer on average extends to a height of 70 m. Control
experiments that were conducted during the model set-up, showed that increasing the number of vertical layers (in the lowest
70 m or throughout the vertical extent of the atmosphere) does not significantly alter the concentrations of pollutants near the
surface, at the station locations, due to mixing within the boundary layer.

Meteorological initial and boundary conditions are provided by the Global Forecast System (GFS) on a horizontal grid
resolution of 0.25° × 0.25°. Time-variant chemical boundary conditions are provided from the global Model for OZone And
Related chemical Tracers (MOZART-4; Emmons et al. (2010)). The MOZART-4 model output datasets are available on a
horizontal grid resolution of 1.9° × 2.5° and interpolated in space every six hours to our model domain and the chemical
species of each mechanism. Biogenic emissions are calculated on-line by the the Model of Emissions of Gases and Aerosols



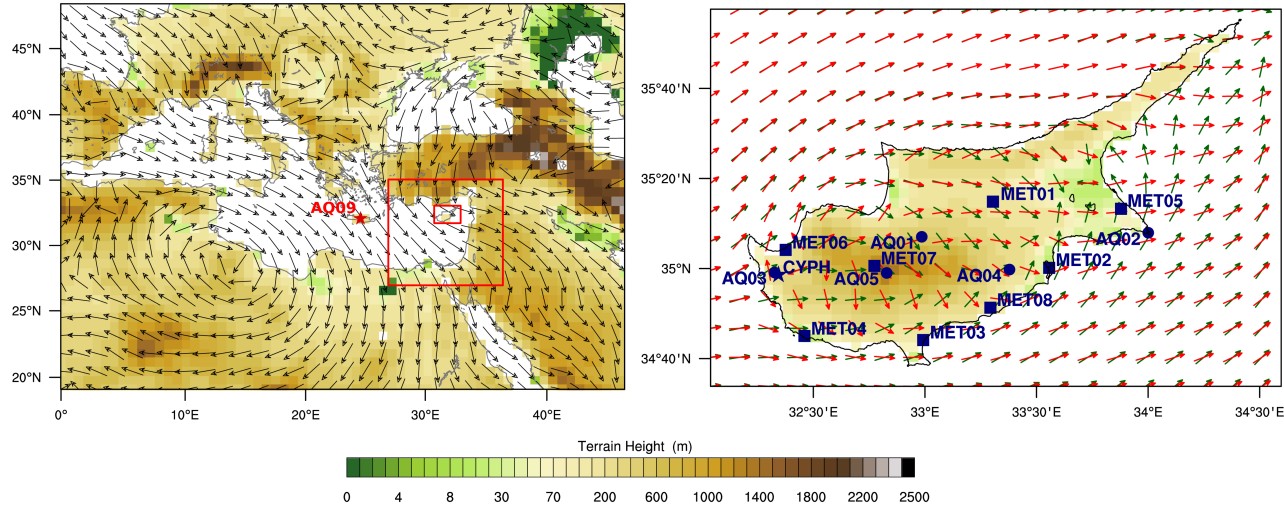

**Figure 1.** Model simulation domains, terrain elevation, mean wind direction at 850 hPa for July 2014, and the location of the Finokalia station (left) and meteorological stations (squares), air pollution stations (circles), CYPHEX Campaign (star), mean nighttime (red vectors), and mean daytime (green vectors) wind direction at 10m (right). Moniroring station details are shown on Table 2

.

from Nature version 2.1 (MEGAN2.1) by Guenther et al. (2012). We use the Air Force Weather Agency (AFWA) dust scheme that was developed based on the Marticorena and Bergametti (1995) dust emission scheme in the Goddard Global Ozone Chemistry Aerosol Radiation and Transport (GOCART) model (Chin et al., 2000). The EDGAR-HTAP (Emission Database for Global Atmospheric Research for Hemispheric Transport of Air Pollution) Version 2, compiled by the European Commission, Joint Research Center (JRC)/Netherlands Environmental Assessment Agency (hhtp://edgar.jrc.ec.europa.eu/htap_v2) is utilized. This dataset includes emissions of gaseous pollutants sulphur dioxide ($SO_2$), $NO_x$, CO, non-methane volatile organic compounds (NMVOCs) and ammonia ($NH_3$) and particulate matter with carbonaceous speciation ($PM_{10}$ , $PM_{2.5}$, black carbon (BC) and organic carbon (OC)) from anthropogenic and biomass burning sectors (Janssens-Maenhout et al., 2012). $PM_{2.5}$ is a subset of $PM_{10}$ and includes BC, OC, $SO_4^{2-}$, $NO_3^-$, crustal material, metal, and other dust particles. The dataset used in this study is available in $0.1^o \times 0.1^o$ emission grid maps for the year 2010 and can be downloaded from the EDGAR JRC website (http://edgar.jrc.ec.europa.eu/) per year, per substance, and per sector.

### 2.3 Observational data

The model output is compared against observational data from a dense station network which spans the island of Cyprus and covers a large variety of monitoring sites, including sea-side and mountainous areas. Specifically, the modelled meteorology is compared against meteorological hourly observations from eight ground stations operated by the Cyprus Department of Meteorology (Figure 1, squares), and meteorological data from the CYPHEX campaign which took place from July $7^{th}$ to July $31^{st}$, 2014, near the Ineia village (Figure 1, star). Modelled pollutant concentrations are compared against observational data



**Table 2.** Air pollution monitoring and meteorological stations.

| Code | Station Name | lat (°) | lon (°) | alt (m) |
|------|-------------|---------|---------|---------|
| CYPH | CYPHEX Campaign | 34.96 | 32.39 | 629 |
| AQ01 | Ayia Marina | 35.04 | 33.06 | 532 |
| AQ02 | Cavo Greco | 34.96 | 34.08 | 17 |
| AQ03 | Ineia Village | 34.96 | 32.38 | 664 |
| AQ04 | Stavrovouni | 34.88 | 33.44 | 512 |
| AQ05 | Troodos | 34.92 | 32.88 | 1745 |
| AQ06 | Finokalia | 35.32 | 25.67 | 250 |
| MET01 | Athalassa | 35.14 | 33.40 | 158 |
| MET02 | Larnaca | 34.87 | 33.62 | 2 |
| MET03 | Limassol | 34.87 | 33.62 | 3 |
| MET04 | Pafos | 34.72 | 32.48 | 10 |
| MET05 | Paralimni | 35.06 | 33.97 | 68 |
| MET06 | Polis | 35.04 | 32.44 | 22 |
| MET07 | Prodromos | 34.95 | 32.83 | 1401 |
| MET08 | Zygi | 34.75 | 33.33 | 40 |

from five background air quality monitoring ground stations operated by the Cyprus Department of Labour Inspection - DLI (Figure 1, circles) and data from the CYPHEX campaign. The Finokalia station in Crete, which is part of the European Monitoring and Evaluation Programme (EMEP) network is used as a reference station to discuss $O_3$ discrepancies on measurements over Cyprus. The frequency of all air pollutant concentrations measurements is hourly, except for $PM_{2.5}$ concentrations mea-
5 surements by the CYPHEX Campaign, which are provided every six hours. The location of the air pollution and meteorology monitoring stations are given in detail in Table 2.

## 3 Results and discussion

### 3.1 Boundary conditions sensitivity tests

Kushta et al. (2017) (accepted for publication) showed that chemical boundary conditions from the MOZART-4 global model
10 have an important effect on the modelled concentrations over the region of study. More specifically, in their study, an important $O_3$ overestimation by the WRF/Chem model was attributed to the effect of chemical boundary conditions. When the $O_3$ inflow from the boundaries was reduced by 30%, model results were closer to observations. Based on these results and the MOZART-4 model evaluation (Emmons et al., 2010), $O_3$ inflow from the global model was reduced by 30% in our study. Figure 2 (left panel) shows the observed and modelled $O_3$ concentrations from the base run using the RMS mechanism and a simulation





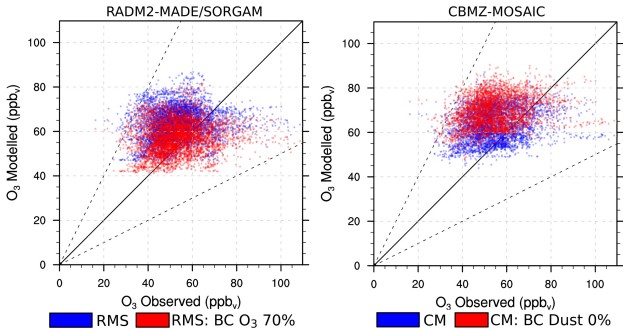

**Figure 2.** $O_3$ observed and modelled concentrations using the RADM2 chemical mechanism with 100% (blue markers) and 70% (red markers) $O_3$ inflow from MOZART-4 (left), and $O_3$ observed and modelled concentrations using the CBMZ chemical mechanism with (blue markers) and without (red markers) dust influx from MOZART-4 (right)

where initial $O_3$ concentrations and $O_3$ inflow from the global model were reduced by 30%. The average NMB decreased from 21% to 9% when $O_3$ from the global model was reduced. Similar results appear at the Finokalia background station, where NMB was reduced from 18% to 7% when $O_3$ inflow from the global model was reduced by 30%. The results for the CM and MM mechanisms are analogous. $O_3$ overestimation due to the effect of boundary conditions from the MOZART-4

model was also reported by Abdallah et al. (2016). In their study, the Polyphemus chemical transport model was found to highly overestimate $O_3$ concentrations over Lebanon when using boundary conditions from the MOZART-4 model. Bossioli et al. (2016) also reported a significant contribution of boundary conditions over the $O_3$ levels in the area. Therefore, $O_3$ from the global model was reduced by 30% for the simulations of this study.

     Since dust is an important parameter of air quality in the region of study and important dust sources are not included in

our outermost domain, dust inflow from the global model was taken into account in our simulations. The effect of dust from the boundaries by the MOZART-4 on the WRF/Chem $PM_{2.5}$ concentrations is examined. In the CM mechanism much higher $PM_{2.5}$ concentrations than those observed occur between July 11 to July 13. To investigate this discrepancy we performed two CM simulations (Figure 3) with (continuous blue line) and without (dashed blue line) dust influx from the boundaries. Incoming dust results in an increase of the order of 19% in $PM_{2.5}$ modelled concentrations during the whole study period. This increase is

more pronounced from July $11^{th}$ to July $13^{th}$ (40%). Dust presence also influences $O_3$ concentrations though aerosol-radiation feedbacks and their impact on photolysis rates. Figure 2 (right panel) shows the observed and modelled $O_3$ concentrations with (blue markers) and without (red markers) dust influx from the global model. The inclusion of dust particles results in a decrease of 10% in modelled $O_3$ concentrations due to changes in solar radiation through aerosol-radiation interactions.

     The comparison of model to station observations is performed using the first free e model layer. The actual altitude of four

stations, located in regions with very complex terrain, was found to differ from the model terrain height (CYPHEX campaign, Ineia village, Troodos air quality monitoring station, and the nearby Prodromos meteorological station). As a test, we performed the comparison using modelled concentrations taking into account the actual altitude of the stations. This resulted only in a





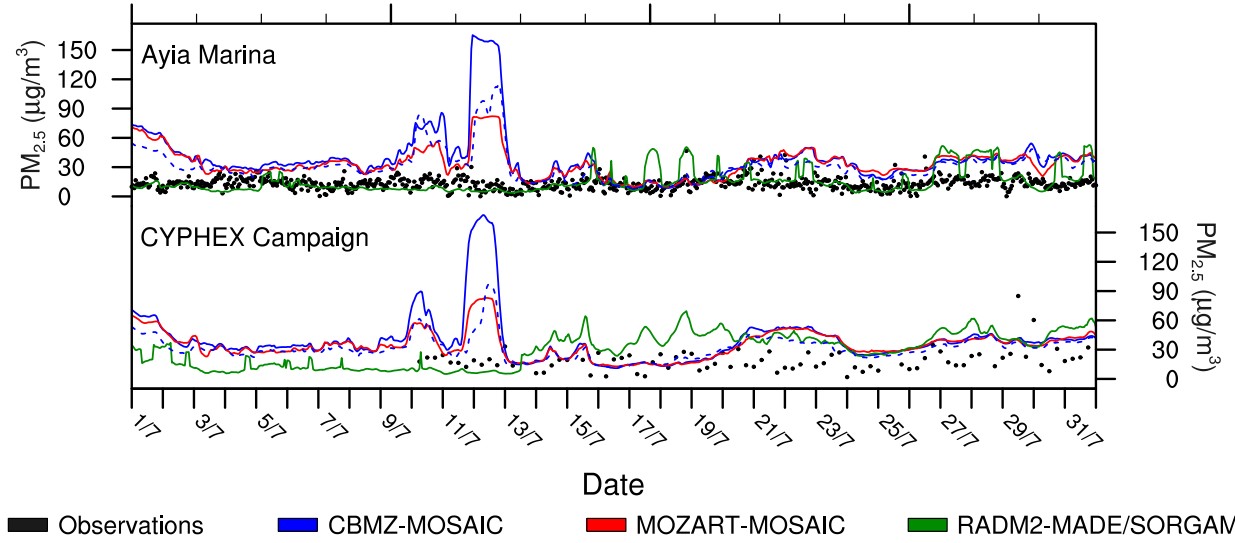

**Figure 3.** Observed (grey markers) and modelled $PM_{2.5}$ concentrations from the CM (blue line), MM (red line), and RMS (green line) mechanisms. The dashed blue line represents the CM simulation without dust influx from the global model. All model simulation output and observations are given in hourly resolution, expect for the CYPHEX Campaign measurements which are provided in 6-hourly resolution

slightly better agreement in the predicted surface pressure at the CYPHEX campaign and the Prodromos station. Results in all other locations were not influenced because of the mixing within the model boundary layer.

### 3.2  Meteorology

We evaluate the model performance regarding basic meteorological parameters. Statistical metrics are derived by comparing the output of the three model simulations to hourly measurements at ground stations. Pearson's Correlation Coefficient (R), Mean Bias (MB), Normalized Mean Bias (NMB), and Root Mean Squared Error (RMSE) for temperature at 2m ($T_{2m}$), wind speed at 10m ($WS_{10m}$), and surface pressure ($P_{surf}$) are shown on Table 3. Modelled $T_{2m}$ is in good agreement with observations (NMB = 2% to 3%) with similar RSME values (2.72 to 2.78 °C) for all three mechanisms. The diurnal cycle of $T_{2m}$ is reproduced at the majority of the stations (R ∼ 0.66). The model though does not capture the $T_{2m}$ diurnal variability at the Larnaca meteorological station (R < 0.20). The station is very close to the sea and the Larnaca Salt Lake that might influence the thermal circulation in the area.

The model tends to overestimate $WS_{10m}$ at the majority of the stations by an average of 1.71 to 1.83 m/s (R ∼ 0.46) for all three mechanisms. Mar et al. (2016) and Zhang et al. (2013) also reported $WS_{10m}$ overpredictions by the WRF model over the Mediterranean. The latter study attributed this model behaviour to the poor representation of surface drag exerted by the unresolved topography (mountains, hills and valleys) and other smaller scale terrain features.



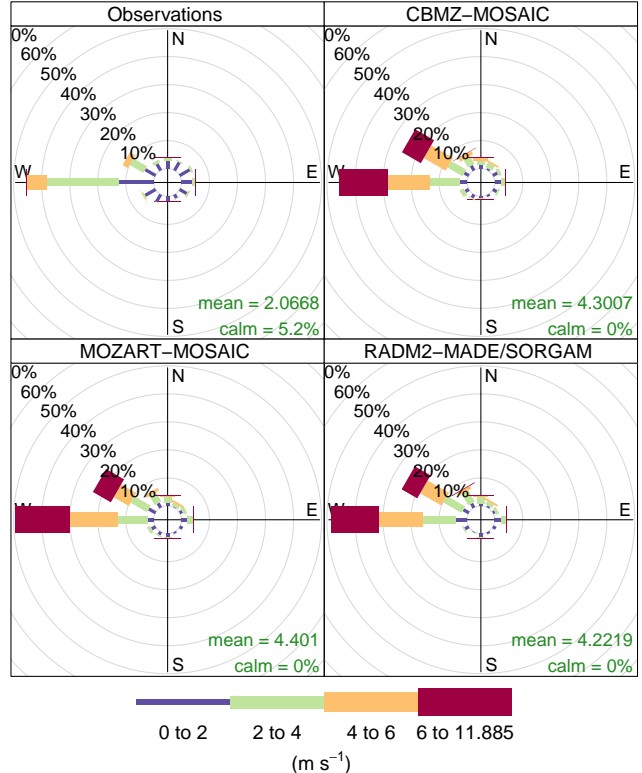

**Figure 4.** Windroses (monthly mean wind speed and direction at 10 m) at the Athalassa Meteorological station (top left), and from the CM (top right), MM (bottom left), and RMS (bottom right) simulations

Local circulation is successfully predicted by the model. Figure 1 (right) shows the average 10 m wind direction from 12:00 to 17:00 LST in green and from 00:00 to 05:00 LST in red color. The model simulates sea breezes during daytime and katabatic winds during the night in agreement with observations. The windroses at the Athalassa station from the observational data (top left panel) and the three simulations show that wind direction is reproduced quite well by the model (Figure 4). The inland dominating wind direction is mainly westerly and nort-westerly with frequency of occurrence 60% and 20% respectively. Similar results appear for the majority of the stations (not shown here). Some discrepancies between model and observations at the Prodromos station are attributed to the complex mountainous topography of the Troodos area. Both model simulations and observational data reveal predominant south-westerly winds at the southern coastline of the island during day and night. The summertime general circulation pattern over the Eastern Mediterranean with predominant northerly and westerly winds, as well as the anticyclonic flow over Western Africa (Figure 1, left panel) are also resembled by the model.

There is very good agreement between observed and modelled $P_{\mathrm{surf}}$ with high hourly correlation coefficients ($R \geq 0.87$) and normalized mean bias of 1%. Some negligible discrepancies exist between the three mechanisms. The differences in the



**Table 3.** Meteorology statistical metrics for the CM, MM, and RMS mechanisms, averaged over all stations, with $O_3$ inflow (reduced by 30%), and dust inflow from the boundaries

| Mechanism | 2m Temperature (°C) | | | | 10m Wind Speed (m/s) | | | | Surface Pressure (hPa) | | | |
|---|---|---|---|---|---|---|---|---|---|---|---|---|
| | R | MB | NMB | RMSE | R | MB | NMB | RMSE | R | MB | NMB | RMSE |
| CM | 0.66 | -0.59 | -0.02 | 2.73 | 0.47 | 1.76 | 1.28 | 2.77 | 0.88 | 3.46 | 0.01 | 12.18 |
| MM | 0.66 | -0.76 | -0.03 | 2.78 | 0.47 | 1.83 | 1.32 | 2.82 | 0.87 | 3.67 | 0.01 | 12.08 |
| RMS | 0.67 | -0.63 | -0.02 | 2.72 | 0.46 | 1.71 | 1.26 | 2.74 | 0.87 | 3.41 | 0.01 | 12.24 |

meteorological components are attributed to the inclusion of the aerosol-radiation feedbacks in the simulations. The model performance regarding aerosol concentrations is discussed later in the paper.



### 3.3 Main gaseous pollutants

Observed average monthly $O_3$ concentrations for July 2014 fall within the climatological averaged summer values given by (Kleanthous et al., 2014). In their study, July monthly means of $O_3$ concentrations over a period of 15 years, were found to be $54.3 \pm 4.7$ ppbV over all stations. The mean observed value during our simulation period at the DLI stations is 52 ppbV.

The mean modelled values vary from 56.2 ppbV (NMB = 9%) in the RMS mechanism to 63 ppbV (NMB = 22%) and 65.2 ppbV (NMB = 23%) in the CM and MM mechanism respectively showing a strong overestimation of the latter two. Figure 5 shows the average $O_3$ ground-level modelled concentrations for the three mechanisms for the outermost domain (Europe - Mediterranean and North Africa). Differences between the three mechanisms are more pronounced over southern Europe and the Mediterranean. Over these regions $O_3$ concentrations predicted by the MM mechanisms are up to 10 ppbV and 20 ppbV

higher compared to the CM and RMS mechanisms respectively.

The CYPHEX campaign station has been excluded from the analysis of $O_3$. This station gives significantly higher average monthly $O_3$ concentration (71.40 ppbV) that deviates from the climatological and observed mean, even though the station of the campaign was located only a few hundred meters away from the Ineia site of DLI (51.93 ppbV). We investigated this deviation by comparing with the mean monthly $O_3$ for July 2014 at the Finokalia station (from the European Monitoring

and Evaluation Programme - EMEP) that reaches 52.43 ppbV. It appears that the CYPHEX measurement site, due to its elevation of about 650m above sea level, was regularly influenced by air from the lower free troposphere with elevated $O_3$ concentrations, which is not representative for boundary layer air. We choose Finokalia because it is located on the island of Crete (approximately 600km west of Ineia) with no pollution sources in between (Figure 1, left panel, red asterisk). Thus Ineia and Finokalia have similar pollution features, being subject to air mass transports from eastern Europe.

Table 4 shows the statistical performance of the three gas-phase and aerosol mechanisms against hourly observations from six ground stations. The CM and MM mechanisms significantly overestimate $O_3$ concentrations with normalized mean biases of 22% and and 23% respectively. A normalized mean bias of 9% appears for the RMS mechanism which corresponds to 4.25 ppbV. This mechanism shows the lowest RMSE (10.97 ppbV) compared to CM and MM (14.79 ppbV and 15.30 ppbV respectively). Our sensitivity tests showed that high $PM_{2.5}$ concentrations affect the $O_3$ concentrations through the aerosol-

radiation feedbacks by altering the radiation budget and as a consequence, the photochemical activity and the concentration of secondary pollutants. Specifically, when the dust inflow from the boundaries for the CM mechanism was not taken into account, $O_3$ concentrations at the stations locations increased by 10%. Since the CM mechanism shows the higher $PM_{2.5}$ concentrations and relatively high $O_3$ concentrations, we conclude that we can rule out the aerosol concentrations and therefore the different aerosol mechanisms, as the responsible factor for the differences in $O_3$ concentrations between the three simulations. Knote

et al. (2014) attributed the differences in pollutants concentrations between these mechanisms to the differences in the treatment of VOCs, since the rate constants for the basic $O_3$ production and loss reactions are similar. Hourly correlation coefficients for $O_3$ are low (less than 0.30) for all three mechanisms. In combination with underestimated $NO_x$ concentrations (NMB varies from -54% to -44%) and low hourly correlation coefficients for $NO_x$ as well, this suggests that nearby anthropogenic emission sources are not represented in the emission inventory used in the simulations.



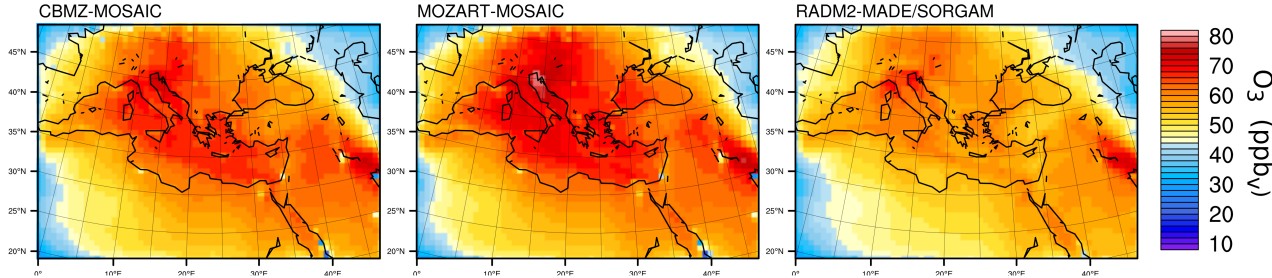

**Figure 5.** $O_3$ **monthly** average ground-level concentrations for CM (left), MM (center), and RMS (right) mechanisms

The hourly modelled and observed $O_3$ concentrations at six background stations over Cyprus are depicted in Figure 6. The three mechanisms show similar behaviour for July $1^{st}$ to July $13^{th}$. However, from July $13^{th}$ until the end of the month $O_3$ concentrations from CM and MM appear slightly higher than RMS. The fact that such differences do not appear for $O_3$ precursors $NO_x$ (Figure 7) indicates that the different behaviour during this period is possibly due to aerosol-radiation interactions and changes in photolysis rates. Similar patterns appear for CO (Figure 8). Due to the long CO atmospheric residence time, these differences can be attributed to long-range transport, and therefore the effect the three different gas-phase chemistry and aerosol mechanisms have on the predicted meteorology.

An abrupt decrease in $O_3$ concentrations in observations from July $11^{th}$ to July $13^{th}$ is also captured by all three mechanisms at all stations except Cavo Greco, which is located in the eastern part of the island (6). This decrease in $O_3$ concentrations is accompanied with a decrease in CO concentrations as demonstrated from the observational data from the CYPHEX Campaign and the WRF/Chem model. No abrupt changes are shown in $NO_x$ concentrations either by the observational data or the model. During this period model results reveal that westerlies account for more than 70% at the stations where decreases in $O_3$ and CO concentrations occur, indicating the transfer of cleaner maritime air from the Mediterranean. In general, wind direction appears to have an important impact on pollutant concentrations over Cyprus. Kleanthous et al. (2014) showed that at the Ayia Marina station northerlies are associated with 3-5% higher $O_3$ concentrations compared to westerlies and southerlies during all seasons. Similar results appear for modelled $O_3$ concentrations at this station. More specifically, northerlies are associated with 4-12% higher $O_3$ concentrations compared to westerlies and southerlies for July 2014.

All three mechanisms tend to underestimate $NO_x$ concentrations at the majority of the stations (NMB varies from -53% to -44%). The Ayia Marina and the Troodos stations are located in a mountainous region which is characterized by steep changes in altitude within short distances. The complexity of the terrain results in inaccuracies in the representation of the local wind circulation by the model which affects the transport of pollutants. This is also supported by the CO model underestimation at the Ayia Marina station. Modelled $NO_x$ concentrations are significantly higher, and in better agreement with observations at the Cavo Greco and Stavrovouni stations. The latter is located a few kilometres to the east of an industrial area, which is represented in the anthropogenic emission inventory used in the simulations. On the other hand, a large nearby highway is not captured by the anthropogenic emission inventory, resulting in peaks in $NO_x$ concentrations from traffic, which are not captured by the model. The Cavo Greco station is located in the eastern part of the island. Model results showed that when





**Table 4.** Gas-phase pollutants and aerosols statistical metrics for the CM, MM, and RMS mechanisms, averaged over all stations, with $O_3$ inflow (reduced by 30%), and dust inflow from the boundaries. The CYPHEX campaign was excluded from the average monthly calculations for $O_3$

| | | CBMZ-MOSAIC | | | | MOZART-MOSAIC | | | | RADM2-MADE/SORGAM | | | |
|---|---|---|---|---|---|---|---|---|---|---|---|---|---|
| | Station | R | MB | NMB | RMSE | R | MB | NMB | RMSE | R | MB | NMB | RMSE |
| | CYPHEX | 0.28 | -6.72 | -0.09 | 13.66 | 0.34 | -6.63 | -0.09 | 13.46 | 0.39 | -15.76 | -0.22 | 19.29 |
| | AQ01 | 0.22 | 12.46 | 0.25 | 16.14 | 0.30 | 13.50 | 0.27 | 16.83 | 0.19 | 6.07 | 0.12 | 11.71 |
| | AQ02 | 0.20 | 15.00 | 0.32 | 17.53 | 0.11 | 15.10 | 0.32 | 18.07 | 0.03 | 7.65 | 0.16 | 12.36 |
| $O_3$ | AQ03 | 0.28 | 12.20 | 0.24 | 15.05 | 0.42 | 12.88 | 0.25 | 15.32 | 0.49 | 4.24 | 0.08 | 8.76 |
| | AQ04 | 0.24 | 12.95 | 0.26 | 15.86 | 0.32 | 13.81 | 0.28 | 16.63 | 0.31 | 7.17 | 0.14 | 11.34 |
| | AQ05 | 0.28 | 2.29 | 0.04 | 9.37 | 0.29 | 3.07 | 0.05 | 9.64 | 0.22 | -3.90 | -0.07 | 9.69 |
| | **Average** | **0.24** | **10.98** | **0.22** | **14.79** | **0.29** | **11.67** | **0.23** | **15.30** | **0.25** | **4.25** | **0.09** | **10.77** |
| | CYPHEX | 0.09 | -0.32 | -0.45 | 5.14 | 0.09 | -0.31 | -0.43 | 5.14 | 0.08 | -0.29 | -0.40 | 5.14 |
| | AQ01 | 0.03 | -0.93 | -0.71 | 1.16 | 0.03 | -0.90 | -0.69 | 1.16 | 0.02 | -0.83 | -0.64 | 1.10 |
| | AQ02 | 0.17 | -0.12 | -0.10 | 1.36 | 0.24 | 0.07 | 0.06 | 1.38 | 0.22 | 0.08 | 0.07 | 1.51 |
| $NO_x$ | AQ03 | 0.29 | -1.35 | -0.77 | 1.76 | 0.30 | -1.33 | -0.76 | 1.74 | 0.42 | -1.32 | -0.75 | 1.71 |
| | AQ04 | -0.10 | -0.96 | -0.42 | 2.39 | -0.08 | -0.75 | -0.32 | 2.47 | -0.09 | -0.61 | -0.27 | 2.49 |
| | AQ05 | 0.06 | -0.54 | -0.72 | 0.65 | 0.06 | -0.55 | -0.73 | 0.66 | -0.16 | -0.47 | -0.63 | 0.62 |
| | **Average** | **0.09** | **-0.70** | **-0.53** | **2.08** | **0.11** | **-0.63** | **-0.48** | **2.09** | **0.08** | **-0.57** | **-0.44** | **2.10** |
| | CYPHEX | 0.03 | -9.85 | -0.10 | 43.82 | 0.02 | -7.04 | -0.07 | 43.28 | 0.06 | -0.62 | -0.01 | 42.26 |
| CO | AQ01 | 0.29 | -21.41 | 0.18 | 27.85 | 0.27 | -18.48 | -0.16 | 25.80 | 0.39 | -13.53 | -0.12 | 21.79 |
| | **Average** | **0.16** | **-15.63** | **-0.14** | **35.84** | **0.15** | **-12.76** | **-0.12** | **34.54** | **0.23** | **-7.08** | **-0.07** | **32.03** |
| | CYPHEX | 0.02 | 19.24 | 0.91 | 43.08 | 0.11 | 13.55 | 0.64 | 29.64 | 0.15 | 15.40 | 0.73 | 30.15 |
| $PM_{2.5}$ | AQ01 | -0.01 | 25.49 | 1.94 | 37.33 | 0.04 | 20.40 | 1.55 | 25.66 | 0.17 | 2.32 | 0.18 | 13.00 |
| | **Average** | **0.01** | **22.37** | **1.43** | **39.71** | **0.08** | **16.98** | **1.10** | **27.65** | **0.16** | **8.86** | **0.46** | **21.58** |
| | CYPHEX | 0.39 | 0.90 | 2.89 | 1.16 | 0.32 | 0.39 | 1.25 | 0.65 | 0.36 | 0.36 | 1.17 | 0.83 |
| $SO_2$ | AQ01 | 0.12 | 0.57 | 1.33 | 0.84 | 0.21 | 0.19 | 0.43 | 0.46 | 0.11 | 0.30 | 0.70 | 0.62 |
| | **Average** | **0.26** | **0.74** | **2.11** | **1.00** | **0.27** | **0.29** | **0.84** | **0.56** | **0.24** | **0.33** | **0.94** | **0.73** |

westerlies occur at this location, $NO_x$ concentrations are significantly higher compared to southerlies for all three mechanisms, indicating that the eastern part of Cyprus is affected by transported pollutants, which are emitted within the island. Average $NO_x$ concentrations during northerlies are also higher compared to southerlies since the station is located south of the main power station of the island (Dekeleia).




**Figure 6.** Observed (grey markers) and modelled O$_3$ concentrations from the CBMZ-MOSAIC (blue line), MOZART-MOSAIC (red line), and RADM2-MADE/SORGAM (green line) mechanisms.





**Figure 7.** Observed (grey markers) and modelled $NO_x$ concentrations from the CBMZ-MOSAIC (blue line), MOZART-MOSAIC (red line), and RADM2-MADE/SORGAM (green line) mechanisms.

## 3.4 Fine Particulate Matter ($PM_{2.5}$)

As shown previously in Figure 3, all three mechanisms overestimate $PM_{2.5}$ concentrations at the Ayia Marina station and the CYPHEX Campaign site. The lowest MB appears for MADE/SORGAM (MB = 8.96 $\mu gm^{-3}$) whereas the MB for MOSAIC is 22.37 $\mu gm^{-3}$ when coupled with CBMZ and 16.98 $\mu gm^{-3}$ when coupled with MOZART.




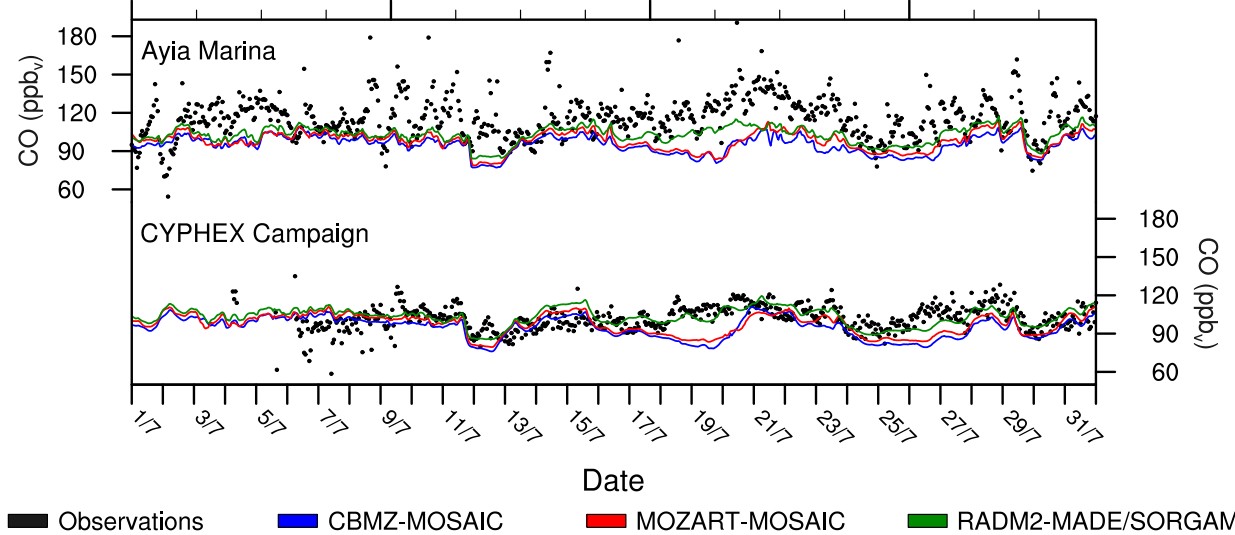

**Figure 8.** Observed (grey markers) and modelled CO concentrations from the CBMZ-MOSAIC (blue line), MOZART-MOSAIC (red line), and RADM2-MADE/SORGAM (green line) mechanisms.

Differences in $PM_{2.5}$ concentrations between the three simulations are more pronounced from July $11^{th}$ to July $13^{th}$. During this period, the average $PM_{2.5}$ concentrations at the Ayia Marina station were 93.95 $\mu gm^{-3}$, 53.81 $\mu gm^{-3}$, and 7.17 $\mu gm^{-3}$ for CM, MM, and RMS respectively, whereas the average $PM_{2.5}$ concentration from observational data was 13.72 $\mu gm^{-3}$. Investigating the inorganic mass in the MOSAIC aerosol mechanism, that represents fine dust particles (Zaveri et al., 2008), we find the overestimation is drve by the dust component.Since observations do not show elevated aerosol levels near the surface from July $11^{th}$ to July $13^{th}$, the dust component has been subtracted from the simulated total $PM_{2.5}$ concentrations (Figure 9). $PM_{2.5}$ concentrations were significantly reduced and are in better agreement with observations for CM and MM simulations, especially from July $11^{th}$ to July $13^{th}$. This indicates that in the MOSAIC aerosol mechanism dust has a great contribution to $PM_{2.5}$ concentrations during this period. On the other hand, smaller differences are shown for the MADE/SORGAM aerosol mechanism. The large difference in $PM_{2.5}$ from mid-month coincides with the time-frame where large discrepancies in gaseous pollutants occur between the three mechanisms, as discussed in Section 3.3. This underlines the importance of the interactions between aerosols and radiation and consequently photolytic reactions and air quality.

In order to better understand individual components of $PM_{2.5}$, and examine differences in behaviour by the aerosol mechanisms, we analyse the aerosol species that dominate the $PM_{2.5}$ concentrations separately. Figure 10 presents the components from observed and modelled sulphate ($SO_4^{2-}$), ammonium ($NH_3^+$), and nitrate ($NO_3^-$) aerosols mass concentrations and elemental carbon concentrations at the Ayia Marina station during the study period. Monthly mean sulphate aerosol concentrations for the three mechanisms vary from 5.14 $\mu gm^{-3}$ to 7.02 $\mu gm^{-3}$ which is close to observed values (5.05 $\mu gm^{-3}$). The lowest monthly mean concentrations are produced by the CM mechanism. This mechanism shows the highest sulphate dioxide ($SO_2$)



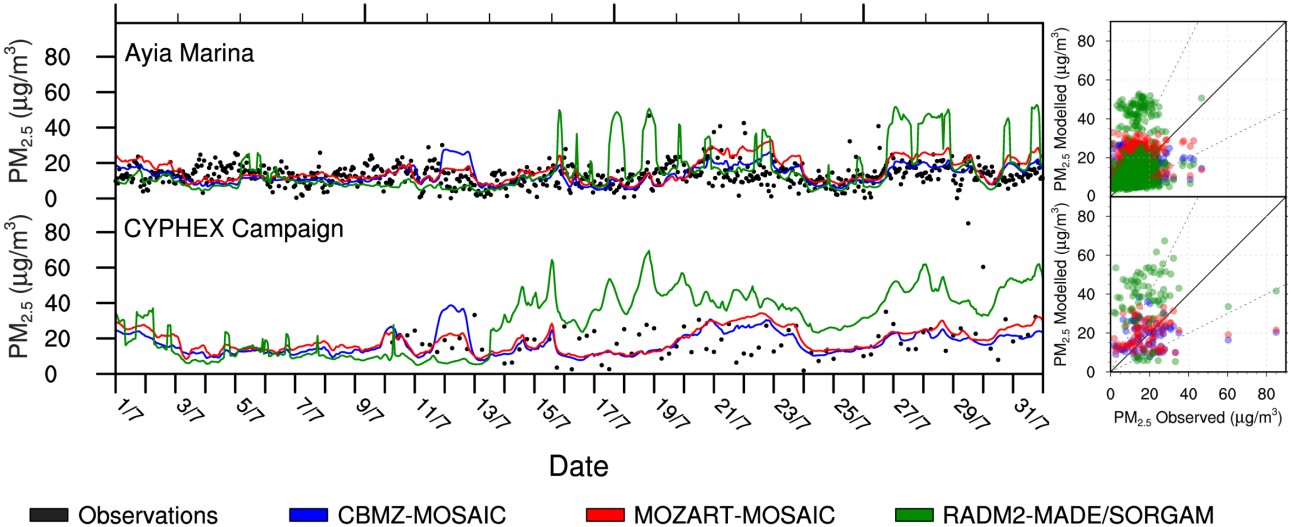

**Figure 9.** Timeseries of observed and modelled $PM_{2.5}$ concentrations from the CM, MM, and RMS mechanisms (left) and scatter plots (right) at the Ayia Marina station and the CYPHEX Campaign. The dust component from the modelled concentrations has been removed from all simulations

concentrations during the whole study period (Figure 11). Since this simulation uses the same aerosol mechanism with the MM simulation, and therefore the same heterogeneous nucleation rates from sulphuric acid ($H_2SO_4$) to sulphate aerosols, the differences between the CM and MM simulations are attributed to the chemical processes that act as sources/sinks of $H_2SO_4$. The RMS mechanism includes the heterogeneous $SO_2$ cloud oxidation (De Brugh et al., 2011; Balzarini et al., 2015) which

results in higher sulphate aerosol concentrations compared to CM and MM.

Elemental carbon is underestimated by all three simulations. The lowest NMB appears for RMS (-34%) followed by CM (-35%). Since the three simulations use the same anthropogenic emission inventory, these differences between RMS and MM can be partially attributed to the different treatment of aerosols by the modal and sectional bin approaches. The MM mechanism highly underestimates EC concentrations due to the absent of anthropogenic emissions in this mechanism.

Ammonium aerosols mean monthly values are close to observed 1.43 $\mu gm^{-3}$ for the CM and MM simulations (1.74 $\mu gm^{-3}$ and 1.87 $\mu gm^{-3}$ respectively), while higher value is shown for RMS (3.24 $\mu gm^{-3}$). Nitrate aerosols are highly overestimated by the RMS mechanism with maximum values and outliers well above the period average. These outliers are due to nitrate aerosols transport from the north, when favoured by wind speed and direction. In contrast, nitrate aerosols are slightly underestimated by CM and MM. These differences lie in the different treatment of the gas-to-particle partitioning from the nitric

acid to ammonium nitrate as a function of humidity from the two aerosol mechanisms used (Balzarini et al., 2015). MADE uses the Mozurkewich (1993) approach and MOSAIC uses the Zaveri et al. (2008) method. The diurnal cycle of compounds that are crucial to night-time chemistry ($NO_3$, $N_2O_5$) vary significantly between the three mechanisms (not shown). The RMS mechanism exhibits three times higher night-time $NO_3$ concentrations than the CM mechanism. This indicates considerable





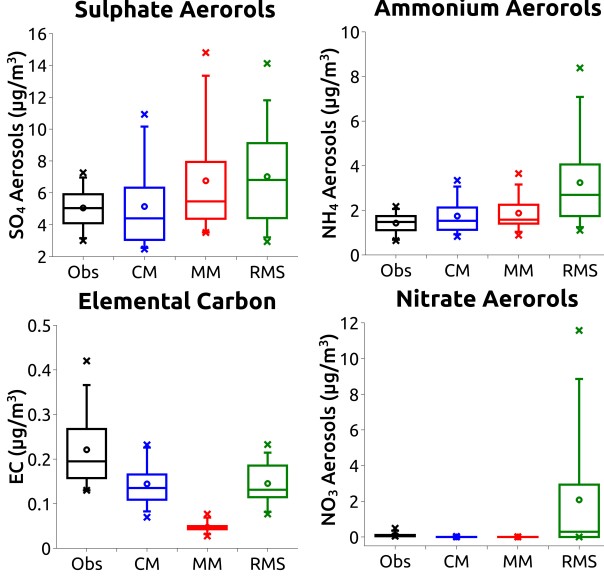

**Figure 10.** Box and whisker plots of observed and modelled sulphate, ammonium, and nitrate aerosols and elemental carbon at the Ayia Marina station for July, 2014

uncertainty in the representation of this important part of tropospheric chemistry that also affects aerosol formation. These results are supported by the findings of Knote et al. (2014) that also report three times higher pan-European averaged $NO_3$ in the RADM2 mechanism compared to CBMZ in the middle of the night-time chemistry cycle.

## 4 Conclusions

We simulated atmospheric gases and aerosols using the WRF/Chem model over the Eastern Mediterranean during the summer. The performance of three gas-phase chemistry and aerosol mechanisms is investigated during the CYPHEX campaign in July 2014. Model output is compared with meteorological and air quality observational data from 14 ground stations. The model reproduces the summertime synoptic wind circulation over the region and the local circulation. It overestimates wind speed at the majority of the stations by an average of 1.71 to 1.83 m/s. Near surface temperature and pressure are reproduced

accurately both in magnitude and diurnal variation. Some discrepancies in modelled and observed meteorological parameters may be attributed to the limited representation of the topography by the model.

Monthly average concentrations of $O_3$ are overestimated by the CM and MM mechanisms by 22 and 23% respectively, whereas a small underestimation is obtained by RMS (9%). Sensitivity tests showed that $PM_{2.5}$ concentrations can affect secondary pollutant formation though aerosol-radiation feedbacks. A decrease of the order of 19% in $PM_{2.5}$ concentrations,

as a result of setting the dust inflow from the global model to zero, resulted in 10% increase in $O_3$ concentrations. The differences in $O_3$ concentrations are attributed to the different treatment of VOCs as per Knote et al. (2014). Hourly correlation





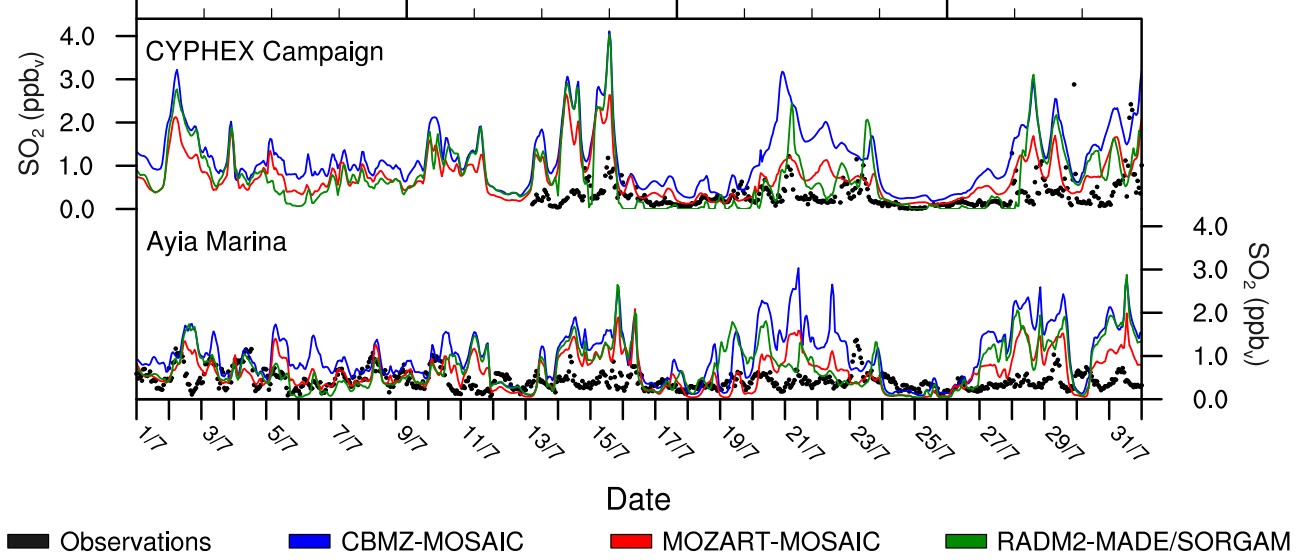

**Figure 11.** Observed (grey markers) and modelled $SO_2$ concentrations from the CBMZ-MOSAIC (blue line), MOZART-MOSAIC (red line), and RADM2-MADE/SORGAM (green line) mechanisms.

coefficients are low for all three mechanisms. $NO_x$ concentrations do not differ between the simulations, with underestimation at the majority of the stations, suggesting that nearby/local anthropogenic emission sources are not well represented in the emission inventory.

Differences in $O_3$ and CO concentrations between the three simulations, during the second half of the simulation period, are attributed to differences in meteorology that derive from the aerosol-radiation interactions. Concurrent abrupt decreases in $O_3$ and CO concentrations (observations and model) during specific days are accompanied with dominance of westerlies carrying clean maritime air from the Mediterranean. Northerlies at the Ayia Marina station are associated with 4-12% higher $O_3$ modelled concentrations compared to westerlies and southerlies for July 2014, which is in good agreement with previous studies.

The terrain complexity in the mountainous areas is the reason for the inaccuracies in the representation of the local wind circulation by the model that affects the transport and vertical mixing of pollutants. As a result, the model performance at these stations (Ayia Marina, Troodos) regarding all pollutants is less satisfactory. On the other hand, the model performance is better in locations with less complex terrain such as the Stavrovouni and Cavo Greco stations. Increased $NO_x$ concentrations at the Cavo Greco station when westerlies occur indicate that the eastern part of Cyprus is affected by emission sources located on the island.

The model skill to reproduce $PM_{2.5}$ concentrations is examined. The MOSAIC aerosol mechanism highly overestimates $PM_{2.5}$ concentrations (NMB $\geq$ 100%). When the dust component is subtracted from the total $PM_{2.5}$ concentrations from all mechanisms there is a better agreement with observations. The RMS mechanism slightly overestimates sulphate and ammonium aerosol at the Ayia Marina station. The CM and MM modelled concentrations of these species are closer to observations



(NMB=2% and 34% respectively). The lowest sulphate aerosol concentrations are produced by the CM mechanism and are accompanied with higher $SO_2$ concentrations. The differences between the two simulations using the MOSAIC aerosol mechanism may be attributed to the chemical processes that act as sources/sinks of $SO_2$. The inclusion of the heterogeneous $SO_2$ cloud oxidation in the RMS mechanism results in higher sulphate aerosol concentrations (NMB = 38%), as described in De

Brugh et al. (2011) and Balzarini et al. (2015). Elemental carbon is underestimated by all three mechanisms indicating lack of emission sources. Differences between the RMS (NMB = -34%) and the CM (NMB = -34%) mechanisms are attributed to to the different approach for the simulation of the aerosol size distribution. Observed and modelled (by CM and MM) nitrate aerosols concentrations at the Ayia Marina site are negligible. RMS simulations yield higher values, probably attributed to nitrate aerosols formation upwind of the measurement site. It is found that key night-time compounds like $NO_3$ and $N_2O_5$

differ significantly between the three mechanisms.

We conclude that all three mechanisms are very sensitive to boundary conditions from the global model for both gas-phase and aerosols pollutants. Care has to be taken, for ozone in particular, which has an important impact on the modelled gas-phase pollutants for all mechanisms. In addition, dust has a great contribution to $PM_{2.5}$ concentrations from the MOSAIC aerosol mechanism, while the corresponding concentrations from CBMZ-MOSAIC were found to be very sensitive to dust from the

boundaries.

*Acknowledgements.* The authors wish to thank the CYprus PHotochemical EXperiment (CYPHEX) Campaign, the Cyprus Department of Meteorology (DoM) and the Cyprus Department of Labour Inspection (DLI), and the EMEP network for providing the observational data used for model evaluation in this study. Plots and diagrams were produced using the NCAR Command Language (NCL) version 6.3.0 (http://www.ncl.ucar.edu/), the openair R package (Carslaw and Ropkins, 2012), and the Qtiplot (http://www.qtiplot.com/). The Computa-

tional resources and support were provided by the European Union Horizon 2020 research and innovation programme VI-SEEM project under grant agreement 675121.



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
