# Peer review of "Air quality modelling in the summer over the Eastern Mediterranean using WRF/Chem: Chemistry and aerosol mechanisms intercomparison"

_Atmospheric Chemistry and Physics, 2017_

## Referee Comment (RC1) · Anonymous Referee #2 · 16 Oct 2017

This manuscript evaluates the online coupled chemistry transport meso-scale model WRF-Chem over eastern Mediterranean with the finest grid resolution (4 km) over Cyprus using different gas phase and aerosol mechanisms. The manuscript has interesting results. I suggest acceptance of the manuscript for publication but after considering a number of comments that follow.

Comments 1) page 2, lines 11-13: It is not actually the Azores High pressure system the anticyclonic center of action that results to the etesian winds in combination with the Asian low. Many researchers underline the differences between the anticyclonic center causing the Etesians and the Azores permanent Anticyclone (Prezerakos 1984;

[Figure]

Tyrlis et al. 2013; Anagnostopoulou et al. 2014) because unlike the Azores anticyclone, the ridge over the Balkans retains its distinct signature up to 500 hPa implying different dynamics involved in its formation. 2) page 2, lines 19-21: There is also a recent study pointing further the role of tropopause folds in summertime tropospheric ozone over the eastern Mediterranean and the Middle East (see Akrtitidis et al., 2016). 3) page 6, line 12: Since the emissions are available at 0.1 deg what is the treatment of the emissions at the finest grid resolution of 4 km; 4) Section 2.3: Please specify the chemical measurements carried out at the air quality stations. 5) page 6, line 30: Please specify if the bias refers to ozone measurements carried out at the five air quality stations of Table 2. 6) page7, line 31: The authors mention ozone overestimation due to the effect of boundary conditions from the global MOZART-4 model (Abadallah et al., 2016). Similar results have been reported in an earlier study implementing chemical boundary from another version of MOZART model showing the importance of time variant chemical boundary conditions for simulated near surface ozone O3 (Akritidis et al., 2013). 7) Table 3 and 4: I think a more detailed description of the Table captions is needed. Specify also the number of the data used to extract the statistical measures. Furthermore I think that the statistical measures should be given for the individual stations, too (even as supplementary material) because one added value of this simulation is the high resolution. If discussion is based on averages of all different stations together, practically the advantage of the high resolution analysis is diminished. 8) How the statistical evaluation measures of this study in Table 4 compare with other similar modelling studies? The correlations based in hourly data are very low for NOx, CO and PM and slightly better but still low for O3. It seems that the day to day variability is not captured adequately. You may check also how model simulates the diurnal variation by comparing the mean modelled diurnal cycles with the observed ones. 9) Page 11, line 15: It is mentioned for the CYPHEX campaign stations that " It appears that the measurement site, due to its elevation of about 650m asl, was regularly influenced from the lower free troposphere with elevated ozone concentrations. . .". However the closest Ineia Village stations is also at a similar altitude but does not show such high values.

Maybe it would be a good idea to compare directly the ozone time series of these two stations. When looking these two ozone time series in Figure 6, I get the impression that there is a co-variability but CYPHEX ozone is constantly higher than Ineia ozone. 10) As the authors mention, Table 4 indicates an overestimation for modelled ozone and an underestimation for modelled NOx. I think it would be interesting to see how the global O3 and NOx values compare with the observations and also how the statistical measures are being modified as we go from the course resolution to the fine resolution of the inner domain. For example, it is important to show if the finer resolution improves the evaluation measures. 11) Page 12, line 5: What do the author mean with "similar patterns appear for CO" in Figure 8? I guess the authors similar to NOx precursors since as I can from Figure 8, CO is underestimated by the model before and after the 13th of July. 12) Since there is a lot of discussion in the manuscript for the role of dust aerosols in the simulation after 11 July (Figures 3 and 9) maybe it would be interesting to see the evolution of simulated dust aerosol optical depth and compare with observed values from available ground based or satellite relevant measurements. This is rather a suggestion than a request for the revision.

Minor comments on the text Page 3, line 21: WRFC/Chem should be WRF/Chem.

---

## Referee Comment (RC2) · Anonymous Referee #1 · 17 Oct 2017

The manuscript has been largely improved compared to the first version. There are still few minor issues I have listed below before the manuscript can be published in ACP.

Section 2.2. How are the EDGAR emissions on 0.1 degree resolution re-gridded into 4 km for the innermost domain?

Section 3.2. Table 3 could also include statistics for all stations so that the discussion for the biases for the pollutant levels can be better addressed referring to Table 3. This can be provided as a supplement. The number if data pairs could be added as an extra information.

Section 3.3. Table 4. The number if data pairs could be added as an extra information.

---

## Author Comment (AC2) · 14 Dec 2017

**Supplement to "Air quality modelling in the summer over the Eastern Mediterranean using WRF/Chem: Chemistry and aerosol mechanisms intercomparison"**

George K. Georgiou[1], Theodoros Christoudias[2], Yiannis Proestos[1], Jonilda Kushta[1], Panos Hadjinicolaou[1], and Jos Lelieveld[3,1]

[1]Energy, Environment and Water Research Center, The Cyprus Institute, Nicosia Cyprus
[2]Computation-based Science and Technology Research Centre (CaSToRC), The Cyprus Institute, Nicosia, Cyprus
[3]Air Chemistry Department, Max Planck Institute for Chemistry, Mainz, Germany

*Correspondence to:* George K. Georgiou (g.georgiou@cyi.ac.cy)

**Table S1.** Pearson's Correlation Coefficient (R), Mean Bias (MB), Normalized Mean Bias (NMB), and Root Mean Squared Error (RMSE) of hourly values of temperature at 2m, wind speed at 10m, and surface pressure for the CBMZ-MOSAIC (CM), MOZART-MOSAIC (MM), and RADM2-MADE/SORGAM (RMS) mechanisms, with $O_3$ inflow (reduced by 30%), and dust inflow from the boundaries. Hourly data availability exceeds 90% at all stations.

| | | CBMZ-MOSAIC | | | | MOZART-MOSAIC | | | | RADM2-MADE/SORGAM | | | |
|---|---|---|---|---|---|---|---|---|---|---|---|---|---|
| | Station | R | MB | NMB | RMSE | R | MB | NMB | RMSE | R | MB | NMB | RMSE |
| **T2** | CYPHEX | 0.70 | 0.65 | 0.03 | 2.34 | 0.69 | 0.49 | 0.02 | 2.36 | 0.68 | 0.69 | 0.03 | 2.42 |
| | MET01 | 0.88 | -0.45 | -0.02 | 2.28 | 0.88 | -0.75 | -0.03 | 2.30 | 0.89 | -0.50 | -0.02 | 2.21 |
| | MET02 | 0.18 | -1.46 | -0.05 | 3.29 | 0.18 | -1.48 | -0.05 | 3.32 | 0.17 | -1.45 | -0.05 | 3.31 |
| | MET03 | 0.65 | -0.75 | -0.03 | 2.84 | 0.65 | -0.86 | -0.03 | 2.89 | 0.67 | -0.84 | -0.03 | 2.80 |
| | MET04 | 0.68 | -1.47 | -0.06 | 2.54 | 0.70 | -1.6 | -0.06 | 2.56 | 0.71 | -1.58 | -0.06 | 2.55 |
| | MET05 | 0.70 | 0.37 | 0.01 | 2.69 | 0.72 | 0.19 | 0.01 | 2.66 | 0.72 | 0.17 | 0.01 | 2.61 |
| | MET06 | 0.74 | -0.84 | -0.03 | 2.64 | 0.76 | -1.05 | -0.04 | 2.65 | 0.79 | -0.86 | -0.03 | 2.46 |
| | MET07 | 0.80 | -1.39 | -0.06 | 2.75 | 0.80 | -1.74 | -0.08 | 2.95 | 0.78 | -1.30 | -0.06 | 2.80 |
| | MET08 | 0.61 | 0.03 | 0.00 | 3.22 | 0.60 | -0.06 | 0.00 | 3.32 | 0.60 | -0.01 | -0.00 | 3.29 |
| | **Average** | **0.66** | **-0.59** | **-0.02** | **2.73** | **0.66** | **-0.76** | **-0.03** | **2.78** | **0.67** | **-0.63** | **-0.02** | **2.72** |
| **PSFC** | CYPHEX | 0.81 | 45.88 | 0.05 | 45.90 | 0.81 | 46.04 | 0.05 | 46.05 | 0.80 | 45.81 | 0.05 | 45.83 |
| | MET01 | 0.87 | 0.10 | 0.00 | 1.26 | 0.87 | 0.31 | 0.00 | 1.29 | 0.86 | 0.07 | 0.00 | 1.35 |
| | MET02 | 0.88 | -2.03 | 0.00 | 2.37 | 0.88 | -1.83 | 0.00 | 2.19 | 0.87 | -2.08 | 0.00 | 2.46 |
| | MET03 | 0.88 | -10.07 | -0.01 | 10.15 | 0.88 | -9.88 | -0.01 | 9.95 | 0.87 | -10.14 | -0.01 | 10.23 |
| | MET04 | 0.91 | -2.19 | 0.00 | 2.44 | 0.91 | -1.96 | 0.00 | 2.23 | 0.90 | -2.25 | 0.00 | 2.54 |
| | MET06 | 0.90 | -10.91 | -0.01 | 10.97 | 0.89 | -10.68 | -0.01 | 10.75 | 0.89 | -10.98 | -0.01 | 11.05 |
| | **Average** | **0.88** | **3.46** | **0.01** | **12.18** | **0.87** | **3.67** | **0.01** | **12.08** | **0.87** | **3.41** | **0.01** | **12.24** |
| **WS$_{10}$** | CYPHEX | 0.36 | 0.05 | 0.02 | 1.63 | 0.36 | 0.09 | 0.03 | 1.66 | 0.35 | -0.01 | 0.00 | 1.63 |
| | MET01 | 0.52 | 2.24 | 1.09 | 2.99 | 0.53 | 2.34 | 1.13 | 3.06 | 0.49 | 2.167 | 1.05 | 2.94 |
| | MET02 | 0.55 | 0.32 | 0.07 | 2.70 | 0.56 | 0.41 | 0.09 | 2.71 | 0.56 | 0.27 | 0.06 | 2.64 |
| | MET03 | 0.62 | 3.37 | 1.51 | 3.82 | 0.61 | 3.49 | 1.57 | 3.93 | 0.58 | 3.33 | 1.50 | 3.80 |
| | MET04 | 0.48 | 0.02 | 0.00 | 1.84 | 0.49 | 0.09 | 0.02 | 1.84 | 0.47 | -0.11 | -0.03 | 1.88 |
| | MET05 | 0.46 | 2.02 | 1.22 | 2.54 | 0.47 | 2.09 | 1.26 | 2.59 | 0.44 | 2.03 | 1.23 | 2.55 |
| | MET06 | 0.46 | 1.51 | 0.93 | 1.96 | 0.46 | 1.56 | 0.96 | 2.02 | 0.48 | 1.54 | 0.94 | 1.96 |
| | MET07 | 0.27 | 2.45 | 2.34 | 3.04 | 0.27 | 2.53 | 2.41 | 3.11 | 0.26 | 2.44 | 2.337 | 3.01 |
| | MET08 | 0.52 | 3.82 | 4.33 | 4.37 | 0.50 | 3.91 | 4.43 | 4.49 | 0.53 | 3.74 | 4.24 | 4.26 |
| | **Average** | **0.47** | **1.76** | **1.28** | **2.77** | **0.47** | **1.83** | **1.32** | **2.82** | **0.46** | **1.71** | **1.26** | **2.74** |

[Figure]

**Figure S1.** Observed $O_3$ concentrations during the CYPHEX Campaign (blue line) and at the Ineia station (red line).

[Figure]

**Figure S2.** Observed (grey line) and modelled $O_3$ monthly mean diurnal cycles from the CBMZ-MOSAIC (blue line), MOZART-MOSAIC (red line), and RADM2-MADE/SORGAM (green line) mechanisms.

[Figure]

**Figure S3.** Observed (grey line) and modelled $NO_x$ monthly mean diurnal cycles from the CBMZ-MOSAIC (blue line), MOZART-MOSAIC (red line), and RADM2-MADE/SORGAM (green line) mechanisms.

**Table S2.** Pearson's Correlation Coefficient (R), Mean Bias (MB), Normalized Mean Bias (NMB), and Root Mean Squared Error (RMSE) of hourly values of $O_3$ and $NO_x$ for the global MOZART-4 model and all three domains of the CBMZ-MOSAIC (CM), MOZART-MOSAIC (MM), and RADM2-MADE/SORGAM (RMS) simulations, averaged over all stations. The CYPHEX campaign was excluded from the mean monthly calculations for $O_3$. Hourly data availability exceeds 90% at all stations except $NO_x$ at the Ineia station ($>75\%$) and the CYPHEX campaign ($>82\%$).

|  | Model/Mechanism | Domain | Resolution (km) | R | MB | NMB | RMSE |
|---|---|---|---|---|---|---|---|
| **O₃** | MOZART-4 | global | ≈215 | 0.27 | 18.93 | 0.37 | 20.96 |
|  | CM | d1 | 80 | 0.18 | 12.18 | 0.24 | 15.65 |
|  |  | d2 | 16 | 0.21 | 11.21 | 0.22 | 14.97 |
|  |  | d3 | 4 | 0.24 | 10.98 | 0.22 | 14.79 |
|  | MM | d1 | 80 | 0.20 | 13.07 | 0.26 | 16.51 |
|  |  | d2 | 16 | 0.27 | 11.56 | 0.23 | 15.14 |
|  |  | d3 | 4 | 0.29 | 11.67 | 0.23 | 15.30 |
|  | RMS | d1 | 80 | 0.19 | 5.08 | 0.10 | 11.27 |
|  |  | d2 | 16 | 0.21 | 4.25 | 0.09 | 10.77 |
|  |  | d3 | 4 | 0.25 | 4.25 | 0.09 | 10.77 |
| **NOₓ** | MOZART-4 | global | ≈215 | 0.09 | -0.70 | -0.43 | 1.82 |
|  | CM | d1 | 80 | -0.03 | -0.50 | -0.28 | 2.01 |
|  |  | d2 | 16 | 0.04 | -0.77 | -0.57 | 2.00 |
|  |  | d3 | 4 | 0.09 | -0.70 | -0.53 | 2.08 |
|  | MM | d1 | 80 | 0.00 | -0.36 | -0.16 | 2.10 |
|  |  | d2 | 16 | 0.07 | -0.69 | -0.51 | 2.02 |
|  |  | d3 | 4 | 0.11 | -0.63 | -0.48 | 2.09 |
|  | RMS | d1 | 80 | -0.01 | -0.41 | -0.21 | 2.08 |
|  |  | d2 | 16 | 0.05 | -0.67 | -0.49 | 2.03 |
|  |  | d3 | 4 | 0.08 | -0.57 | -0.44 | 2.10 |

[Figure]

**Figure S4.** Comparison of observed and modelled $O_3$ (a, first row) and $NO_x$ (b, secodn row) concentrations at all stations from the CBMZ-MOSAIC (left column), MOZART-MOSAIC (central column), and RADM2-MADE/SORGAM (right column) mechanisms for the 80km (blue color), 16km (red color), and 4km (green color) domain of the simulations. The corresponding concentrations from the global MOZART-4 (215km) are also shown (magenta color).

---

## Author Response (AR1)

Dear Editor,

We would like to thank the reviewers for the constructive comments and suggestions on how to improve the manuscript. Please find our replies to all comments below.

**Reviewer Report #1**

The manuscript has been largely improved compared to the first version. There are still few minor issues I have listed below before the manuscript can be published in ACP.

**Specific Comments:**

1. **Section 2.2. How are the EDGAR emissions on 0.1 degree resolution re-gridded into 4 km for the innermost domain?**

   *Anthropogenic emissions were interpolated in space and time to produce daily emissions using the anthro_emiss utility (https://www2.acom.ucar.edu/wrf-chem/wrf-chem-tools-community) which creates WRF gridded anthropogenic emission files from lat/lon gridded anthropogenic emission files.*

2. **Section 3.2. Table 3 could also include statistics for all stations so that the discussion for the biases for the pollutant levels can be better addressed referring to Table 3. This can be provided as a supplement. The number of data pairs could be added as an extra information.**

   *Statistical measures for individual stations for meteorology (Table 3) have been added in Table S1 in the Supplement. Data availability is noted in the caption.*

3. **Section 3.3. Table 4. The number of data pairs could be added as an extra information.**

   *Data availability is noted in the caption.*

**Reviewer Report #2**

This manuscript evaluates the online coupled chemistry transport meso-scale model WRF-Chem over eastern Mediterranean with the finest grid resolution (4 km) over Cyprus using different gas phase and aerosol mechanisms. The manuscript has interesting results. I suggest acceptance of the manuscript for publication but after considering a number of comments that follow.

**Specific Comments:**

1. **page 2, lines 11-13: It is not actually the Azores High pressure system the anticyclonic center of action that results to the etesian winds in combination with the Asian low. Many researchers underline the differences between the anticyclonic center causing the Etesians and the Azores permanent Anticyclone (Prezerakos 1984; Tyrlis et al. 2013; Anagnostopoulou et al. 2014) because unlike the Azores anticyclone, the ridge over the Balkans retains its distinct signature up to 500 hPa implying different dynamics involved in its formation.**

    *Changed and relevant references added.*

2. **page 2, lines 19-21: There is also a recent study pointing further the role of tropopause folds in summertime tropospheric ozone over the eastern Mediterranean and the Middle East (see Akrtitidis et al., 2016).**

    *We have added a reference to the study in our discussion about the tropopause folds in summertime ozone over the eastern Mediterranean and the Middle East.*

3. **page 6, line 12: Since the emissions are available at 0.1 deg what is the treatment of the emissions at the finest grid resolution of 4 km;**

    *Anthropogenic emissions were interpolated in space and time to produce daily emissions using the anthro_emiss utility (https://www2.acom.ucar.edu/wrf-chem/wrf-chem-tools-community) which creates WRF gridded anthropogenic emission files from lat/lon gridded anthropogenic emission files.*

4. **Section 2.3: Please specify the chemical measurements carried out at the air quality stations.**

   *The measurements carried out at each station are now shown on Table 2.*

5. **page 6, line 30: Please specify if the bias refers to ozone measurements carried out at the five air quality stations of Table 2.**

   *We clarified in the manuscript that the bias refers to the five air quality stations of Table 2 and the CYPHEX campaign.*

6. **page7, line 31: The authors mention ozone overestimation due to the effect of boundary conditions from the global MOZART-4 model (Abadallah et al., 2016). Similar results have been reported in an earlier study implementing chemical boundary from another version of MOZART model showing the importance of time variant chemical boundary conditions for simulated near surface ozone O3 (Akritidis et al., 2013).**

   *We have added a reference to the study in our discussion about the effects of boundary conditions on regional air quality modeling.*

7. **Table 3 and 4: I think a more detailed description of the Table captions is needed. Specify also the number of the data used to extract the statistical measures. Furthermore I think that the statistical measures should be given for the individual stations, too (even as supplementary material) because one added value of this simulation is the high resolution. If discussion is based on averages of all different stations together, practically the advantage of the high resolution analysis is diminished.**

   *A more detailed description has been added to the Table captions. Captions also contain the availability of data used to extract the statistical values at each station. Statistical measures for individual stations for meteorology have been added in Table S1 in the Supplement.*

8. **How the statistical evaluation measures of this study in Table 4 compare with other similar modelling studies? The correlations based in hourly data are very low for NOx, CO and PM and slightly better but still low for O3. It seems that the day to day variability is not captured adequately. You may check also how model simulates the diurnal variation by comparing the mean modelled diurnal cycles with the observed ones.**

*Air quality modelling studies over the Eastern Mediterranean in the literature mainly focus on $O_3$. During the second phase of the Air Quality Model Evaluation International Initiative (AQMEII), the majority of the modelling groups using the RMS and CM mechanisms with the WRF/Chem model also reported $O_3$ concentrations overestimation over the Eastern Mediterranean (Im et al., 2015). On the contrary, Mar et al., (2016) reported an underestimation of about 5ppbv in summertime $O_3$ concentrations WRF/Chem model using the RMS mechanism. Hourly Correlation Coefficients for $O_3$ are comparable to Mar et al., (2016) (R ≈ 0.2 at the Ayia Marina station during summertime). The comparison of observed diurnal cycles for $O_3$ and $NO_x$ has been added in the supplement (Figures S2 and S3 respectively). The fact that no pronounced diurnal cycle is shown on observational data either (except $NO_x$ at the Stavrovouni station which is located close to the highway) indicates that long-range transport is an important aspect of air quality over Cyprus.*

9. **Page 11, line 15: It is mentioned for the CYPHEX campaign stations that " It appears that the measurement site, due to its elevation of about 650m asl, was regularly influenced from the lower free troposphere with elevated ozone concentrations. . ." . However the closest Ineia Village stations is also at a similar altitude but does not show such high values. Maybe it would be a good idea to compare directly the ozone time series of these two stations. When looking these two ozone time series in Figure 6, I get the impression that there is a co-variability but CYPHEX ozone is constantly higher than Ineia ozone.**

*We agree with the observation by the reviewer and the sentence is now removed. Regarding the ozone concentrations at the CYPHEX Campaign, they are indeed constantly higher than at the Ineia station. A direct comparison between these two measurements sites has been added in the supplement (Figure S1).*

10. **As the authors mention, Table 4 indicates an overestimation for modelled ozone and an underestimation for modelled NOx. I think it would be interesting to see how the global O3 and NOx values compare with the observations and also how the statistical measures are being modified as we go from the coarse resolution to the fine resolution of the inner domain. For example, it is important to show if the finer resolution improves the evaluation measures.**

*We have added the scatter plots of the observed and modelled $O_3$ and $NO_x$ concentrations from the global MOZART-4 model and the three domains of the WRF/Chem model for each simulation in the supplement (Figure S4) . The corresponding statistical measures are also shown on Table S2 in the supplement. The global model overall simulates NOx concentrations more accurately. This is attributed to the emission inventory used, as stated in the manuscript. There is a substantial improvement on simulated $O_3$ concentrations when moving from the global model to the outermost domain of the WRF/Chem model (NMB reduced by 26% for the RMS mechanism and by 11% and 13% for the MM and CM mechanisms respectively). A further improvement of the order of 3% is shown when moving from the coarse to the finer WRF/Chem domain on $O_3$ statistical measures.*

11. **Page 12, line 5: What do the author mean with "similar patterns appear for CO" in Figure 8? I guess the authors similar to NOx precursors since as I can from Figure 8, CO is underestimated by the model before and after the 13th of July.**

*We clarified in the manuscript that we mean that the three mechanisms show similar behaviour from July 1st to July 13th for CO as well, but the RMS mechanism gives higher CO concentrations for the rest of the simulation period.*

12. **Since there is a lot of discussion in the manuscript for the role of dust aerosols in the simulation after 11 July (Figures 3 and 9) maybe it would be interesting to see the evolution of simulated dust aerosol optical depth and compare with observed values from available ground based or satellite relevant measurements. This is rather a suggestion than a request for the revision.**

*This is a very good suggestion by the referee. We are planning a follow-up paper focusing on aerosols in which simulated aerosols and their properties over the region of study will be examined in more detail and over a longer time span to capture seasonality.*